# Intelligent Sensing of Thermal Error of CNC Machine Tool Spindle Based on Multi-Source Information Fusion

**DOI:** 10.3390/s24113614

**Published:** 2024-06-03

**Authors:** Zeqing Yang, Beibei Liu, Yanrui Zhang, Yingshu Chen, Hongwei Zhao, Guofeng Zhang, Wei Yi, Zonghua Zhang

**Affiliations:** 1School of Mechanical Engineering, Hebei University of Technology, Tianjin 300130, China; yangzeqing@hebut.edu.cn (Z.Y.); 15031244682@163.com (B.L.); chenyingshu@hebut.edu.cn (Y.C.); zhangzonghua@hebut.edu.cn (Z.Z.); 2Key Laboratory of Hebei Province on Scale-Span Intelligent Equipment Technology, Hebei University of Technology, Tianjin 300130, China; 3State Key Laboratory for Manufacturing Systems Engineering, Xi’an Jiaotong University, Xi’an 710049, China; 17622351275@163.com (H.Z.); cw211512@foxmail.com (G.Z.); 4State Key Laboratory of Strength and Structural Integrity, Aircraft Strength Research Institute of China, Xi’an 710065, China; wei853193@gmail.com

**Keywords:** spindle, thermal-error modeling, multi-source information fusion, intelligent perception, machine tool

## Abstract

Aiming at the shortcomings of single-sensor sensing information characterization ability, which is easily interfered with by external environmental factors, a method of intelligent perception is proposed in this paper. This method integrates multi-source and multi-level information, including spindle temperature field, spindle thermal deformation, operating parameters, and motor current. Firstly, the internal and external thermal-error-related signals of the spindle system are collected by sensors, and the feature parameters are extracted; then, the radial basis function (RBF) neural network is utilized to realize the preliminary integration of the feature parameters because of the advantages of the RBF neural network, which offers strong multi-dimensional solid nonlinear mapping ability and generalization ability. Thermal-error decision values are then generated by a weighted fusion of different pieces of evidence by considering uncertain information from multiple sources. The spindle thermal-error sensing experiment was based on the spindle system of the VMC850 (Yunnan Machine Tool Group Co., LTD, Yunnan, China) vertical machining center of the Yunnan Machine Tool Factory. Experiments were designed for thermal-error sensing of the spindle under constant speed (2000 r/min and 4000 r/min), standard variable speed, and stepped variable speed conditions. The experiment’s results show that the prediction accuracy of the intelligent-sensing model with multi-source information fusion can reach 98.1%, 99.3%, 98.6%, and 98.8% under the above working conditions, respectively. The intelligent-perception model proposed in this paper has higher accuracy and lower residual error than the traditional BP neural network perception and wavelet neural network models. The research in this paper provides a theoretical basis for the operation, maintenance management, and performance optimization of machine tool spindle systems.

## 1. Introduction

The spindle system plays a crucial role in machine tools, and its stability and accuracy directly affect the machining quality. Under complex working conditions, the uneven distribution of the temperature field of the spindle system may lead to unpredictable thermal deformation, which, in turn, affects the machining accuracy. Therefore, studying intelligent-sensing methods for spindle thermal errors is crucial for improving the performance and intelligence of CNC machine tools. The ability of the spindle system to independently perceive its own state and environmental conditions can allow the system to realize real-time monitoring and adjustment of the machining process, thereby improving machining accuracy and stability. The research on this intelligent-perception method not only helps to improve the machining quality of the machine tool but also enhances the intelligent level of operation and maintenance management of the machine tool, making it more adaptable to the complex and changing machining environment and its needs [1,2].

The research on intelligent sensing of thermal error in machine tool spindles mainly has two aspects: thermal-error signal analysis and thermal-error modeling. Thermal-error signal analysis mainly collects the relevant signals in the machine tool and the surrounding machining environment through temperature sensors, displacement sensors, etc. It performs signal analysis and feature extraction on them. Brecher, C. et al. [3] used the unscented Kalman filter (UKF) to estimate machine tool kinematic error model parameters. The kinematic error model of a machine tool contains the time-varying errors, both geometric and thermal. The researchers used an unscented Kalman filter to fuse three-dimensional probe data with a low sampling rate, three-dimensional probe data with a high sampling rate, and comprehensive deformation sensor data with a high sampling rate for the real-time calibration of thermal-error models. This reduces the impact of modeling errors caused by nonlinearity and measurement noise and improves the machining accuracy and stability of the machine. Guo et al. [4] proposed a static thermal deformation modelling method (ST-CLSTM) for machine tools based on a spatiotemporal correlation hybrid CNN-LSTM. They used a convolutional neural network (CNN) to extract temperature features and construct the dataset and a long short-term memory (LSTM) network to capture the temperature change features, considering the sequential nature of the temperature data. The experiment verifies that the model has higher prediction accuracy than the traditional model and solves the problem of temperature-sensitive point selection in thermal-error modeling. Jia et al. [5] constructed a thermal-error prediction model using a one-dimensional convolutional neural network-gated recurrent unit (1DCNN-GRU-Attention). The convolution module is used to replace the traditional temperature-sensitive point selection method. The experiment’s results show that the prediction accuracy of the proposed model is 81.53% under multi-coupled factors. The root-mean-square error (RMSE) is 40% lower than that of the traditional method.

Regarding thermal-error modeling, there are presently mainly thermal-error modeling methods based on heat transfer theory, and polynomial fitting or neural network modeling methods based on experimental data [6]. The thermal-error modeling method based on heat transfer theory is mainly based on the energy conservation equation of heat conduction–convection–thermal radiation used to solve the temperature field and the corresponding displacement field of the machine tool spindle or key components. The classical methods are the centralized mass method and the finite element method [7]. The centralized mass method simplifies the geometry and material distribution of the analyzed object, and by reasonably selecting the location and mass value of the concentrated mass points, connecting each of them with each other using thermal resistance, and establishing the energy conservation equation, a thermal-error model can be obtained in order to predict the characteristics and response of the structure [8].

Kim et al. [9] used the centralized mass method to model the thermal error of the ball screw feed drive system to calculate the temperature distribution and thermal deformation of the ball screw feed drive system. Huang et al. [10], who used the centralized mass method to model the thermal errors of tension rods and bending beams, investigated the relationships between thermal deformation and temperature and heat. In this case, the thermal-error model of the spindle was established by using the thermoelastic mechanics theory and the lumped heat capacity method, and the average fitting accuracy of the model reached 91.3%. The finite element analysis method uses finite element software to model the structure and material properties of the machine tool. Then, it analyzes the thermal error of the machine tool under different operating conditions [11]. Wu et al. [12] used the finite element method to analyze the thermal characteristics of a ball screw feed drive system under long-term operating conditions. By estimating the heat source intensity, based on the temperature profile, through inverse analysis, the primitive domain of the ball screw is divided into multiple units so that the discrete system is equivalently replaced by a continuous system. The temperature distribution is then converted into transient heat transfer in a non-deforming medium, and the thermal expansion of the ball screw is simulated based on the calculated heat flux. Yang et al. [13] have numerically simulated the thermal expansion process of a high-speed motorized spindle under normal operating conditions using a transient thermal–structural coupled finite element analysis method. The finite element prediction results were also compared with the measured temperatures and deformations, and it was found that the thermal model can be used to predict the transient thermal characteristics under various operating conditions. Ma et al. [14] developed a three-dimensional finite element model, considering thermal contact resistance and bearing stiffness, for transient thermal–structural coupling analysis of a high-speed electrical spindle. They verified the validity of the model through thermal balance experiments.

Thermal-error modeling methods based on heat transfer theory generally involve complex mathematical and physical equations, which may require significant computational resources and time, to solve complex structures. The centralized mass method and the finite element method usually require some approximations to reduce the computational complexity, but these approximations, to a certain extent, affect the accuracy of the analysis results, which need to be evaluated and verified according to the actual situation in the specific application.

A polynomial fitting or neural network modeling method based on experimental data is used to construct a correlation model using internal and external information from the machine, such as temperature rise at sensitive points and machine speed obtained from thermal-error experiments, as inputs, and thermal drift as output. Among these methods, the polynomial fitting method usually uses a polynomial function to fit the thermal-error data of a machine tool in order to model and predict it. In contrast, the neural network method trains a neural network model with a large amount of data, thus realizing the modeling and prediction of the thermal error of the machine tool. These classical methods have been widely used and studied in practical engineering.

Gowda Chethana, R. et al. [15] used multiple linear regression methods to develop a prediction model for thermal errors in CNC machine tools. They used the experimentally measured diameter deviation as the dependent variable and the temperature data as the independent variable. They obtained the regression coefficients in multiple linear regressions using the least-squares method to determine the deviation between the tool point and the workpiece. The experiment’s results show that the method effectively predicts the radial deviation of CNC machine tools. Zhao et al. [16] proposed a three-dimensional thermal-error analysis method based on rotation error vector and translation error vector, obtained six vectors of thermal error in the three-dimensional space of the spindle through the testing experiments, and used the thermal-error compensation technique of space coordinate transformation parameters to verify the machining of S samples before and after the compensation for the thermal error. The machining accuracy of the parts was improved by 34.1%, which laid a theoretical foundation for the detection of and compensation for thermal error in asymmetric spindles in the same kind of high-torque CNC machine tools. Li [17] and others established a least-squares support vector machine (LSSVM) prediction model optimized by Aquila Optimizer (AO), and the experiment’s results showed that the prediction accuracy of the AO-LSSVM prediction model for the thermal error of the electrical spindle can reach 94%, and it has a good stability and generalization ability. Huang et al. [18] introduced a genetic algorithm to optimize the initial weights and thresholds of the traditional back-propagation neural network. They used the combination of genetic algorithm and neural network in the thermal-error prediction of high-speed spindles, which showed advantages in solving the global minimum search problem quickly, compared with the traditional back-propagation neural network model. Lee et al. [19] applied fuzzy logic decision to thermal-error modeling, and many other scholars [20,21,22] have applied a gray model and an artificial neural network to spindle thermal-error prediction, an approach which has fewer learning samples and avoids the loss of information in a single modeling approach. All of these methods reflect the thermal error by modeling the correlation between the temperature-sensitive points and the thermal deformation, so selecting the temperature-sensitive points is particularly important.

The methods mainly used for selecting temperature-sensitive points are the empirical correlation coefficient and the cluster analysis [23]. The empirical analysis method empirically analyses the components in which thermal deformation occurs and uses the temperature-sensitive points associated with them as model inputs. Among these efforts, Wu et al. [24] used three temperature measuring points, associated with spindle speed, spindle movement, and coolant system, as temperature-sensitive points; Yang et al. [25] used measuring points related to the spindle base, X-axis screw, and spindle column as the temperature-sensitive points, according to empirical statistics; the correlation coefficient method involves screening the temperature measuring points by describing the correlation coefficient between the temperature field and the thermal error to determine its temperature-sensitive point. Using this, Guo et al. [26] used the correlation coefficient to classify the 12 groups of temperature measuring point data. A group of data was selected in each category to reflect the temperature information in the group. Three groups of temperature-sensitive-point data were obtained as inputs for the model after screening. Liu et al. [27] used the correlation coefficient method to evaluate the correlation between temperature-sensitive points and thermal errors. They found that this method has advantages in selecting temperature-sensitive points that remain stably correlated with thermal errors over time. The cluster analysis method of screening temperature-sensitive points generally uses a clustering algorithm to group temperature data, calculate the distance between data points and the clustering centers, and group data points with similar temperatures into one category. Fu et al. [28] used correlation analysis and K-means clustering to select combinations of global temperature sensitivities for machine tools. Liu et al. [29] used a fuzzy clustering algorithm combined with average influence values to optimize the temperature collection points, which ensured the robustness of the model by classifying the variables and prompting the selection of typical variables to reduce the inputs. Hu [30] used fuzzy C-mean (FCM) clustering and correlation analysis to select temperature-sensitive points in the thermal-error modeling and introduced the Dunn index to determine the optimal number of clusters, a tactic which can effectively suppress the multicollinearity problem between temperature measuring points. Li et al. [31] used the fuzzy C-mean clustering algorithm to screen the temperature-sensitive points and then used the Pearson correlation coefficient to improve the covariance and correlation between the temperature variables; the covariance and correlation problems between the temperature variables were effectively weakened.

In summary, most existing studies only consider the spindle temperature field’s influence on the spindle’s thermal error. In contrast, the spindle speed, spindle load, cooling system, spindle structure, ambient temperature, lubrication, and the thermal conductivity of processing materials, and so on, will affect the thermal error of the spindle system in the actual working process, so it is necessary to analyze the internal and external information from the machine tool in a comprehensive perception of the tool and the use of multi-source information fusion can help to obtain a more accurate thermal-error model.

The rest of this paper is structured as follows: Section 2 firstly briefly analyzes the generation mechanism of the thermal error of the CNC machine tool spindle and then systematically introduces the proposed intelligent-sensing method and model of spindle thermal error based on multi-source information fusion and describes in detail the multi-source information feature extraction and information fusion algorithm in the model. Section 3 describes the experimental design testing thermal-error sensing for spindles. Then, the experiment’s results are analyzed and discussed in Section 4, and the proposed intelligent thermal-error sensing method based on multi-source information fusion is compared and analyzed with the existing algorithms. Finally, conclusions are drawn in Section 5.

## 2. Intelligent-Sensing Method of Thermal Error

### 2.1. Mechanism Analysis of Thermal Error of CNC Machine Tool Spindle

During the machining process, CNC machine tools are affected by internal and external heat sources such as cutting heat, friction heat, and the surrounding environment. The heat generated {Q} is transferred to the machine components by radiation, convection, and conduction. This can result in an uneven distribution of the temperature field {ϕ} and the thermal deformation {u} of the machine tool due to the incomplete symmetry of the structure of the machine tool, the different materials of the internal components, and the differences in the degree of heat dissipation on the surface of the machine tool. This causes the CNC machine to produce a change in the relative positions of the components compared to the standard steady-state condition, which ultimately leads to a relative displacement {δ} between the workpiece and the tool, affecting the machining accuracy, as shown in Figure 1.

As a critical core component of CNC machine tools in high-speed operation, the spindle will be subject to friction, cutting heat, and other factors, resulting in a rise in the spindle’s local temperature and causing thermal deformation. Meanwhile, the spindle is affected by high temperatures during the working process, and the local area will be affected by thermal stress, which will change the shape of the spindle and cause thermal error. In addition, with the complexity and variability of the machining environment and working conditions, the spindle in the operation process or machining process produces unpredictable thermal errors, so it is vital to carry out intelligent sensing of the spindle thermal error of the machine tool under complex working conditions. In the actual operation of the machine tool, machine settings, staff operation, maintenance, calibration, and other aspects of these factors may have impacts on the thermal error. However, the impact on the thermal error is relatively small, so in order to simplify the model analysis, these factors can be ignored.

Existing thermal-error sensing models generally model the correlation between temperature-sensitive points and thermal deformation. However, the selection of the sensitive points, given that the thermal-error model has been established using a single sensor, has certain limitations. These points cannot accurately reflect the correlation of temperature field and thermal deformation under different working conditions. In order to improve the robustness of the thermal-error sensing model, the factors affecting the thermal error of the spindle system are analyzed, and it is found that the main signals related to the thermal error of the spindle are the spindle temperature field, the spindle working-condition parameters, and the motor current signal. In addition, elements of working-condition information such as the spindle reach, thermal balance time, spindle speed, and size of the cutting force directly affect the thermal deformation of the spindle; when the spindle is at a high speed, the friction and cutting heat will increase, resulting in rises in the spindle’s temperature, increasing the possibility of thermal error. The spindle motor current signal reflects the spindle load condition and working status, and the increase in cutting load will increase the spindle force, generate more heat, and aggravate the thermal error of the spindle system.

Therefore, the work of utilizing sensors to collect multi-source thermal information associated with both the machine tool spindle itself and the machining environment, and analyzing, processing, and fusing it in real-time to construct a thermal-error model is indeed critical to improving the performance and accuracy of the machine tool. Such an intelligent-sensing system allows the machine tool to know, in real time, its own thermal state and the influence of the surrounding environment, allowing it to make timely adjustments and optimizations during the machining process to improve the quality and efficiency of its machining.

### 2.2. CNC Machine Tool Spindle Thermal-Error Sensing Methods

Thermal-error sensing of machine tool spindles generally requires thermal performance tests under different operating conditions to obtain temperature rise data and thermal deformation data at measurement points. Then, using the test data, with the corresponding method for the optimization of the selection of each temperature measurement point, the experimental thermal deformation data and optimized measurement point data are used to establish the spindle system’s temperature-sensitive points and determine the thermal deformation using the correlation model. This employs a large number of samples to obtain the parameters of the correlation model, and then relies on the measured temperature rise data for thermal-error sensing prediction.

Referring to the structural characteristics of the vertical machining center spindle and the thermal deformation mechanism, this paper proposes an intelligent-sensing method architecture for spindle thermal error based on multi-source information fusion, as shown in Figure 2. This consists of three main layers: the perception layer, the analysis layer, and the reasoning decision-making layer. The perception layer senses the internal and external signals of the machine tool spindle system by arranging a number of sensors, including an infrared thermal imager (Japan Avionics Co.,Ltd. Yokohama, Japan) and eddy current displacement sensors, and inputs them into the PC to complete the acquisition of signals. The analysis layer prepares the multi-source information by filtering, denoising, and utilizing other pre-processing and feature extraction techniques to achieve the screening of information, in order to obtain the spindle temperature, motor current, and spindle operating conditions, as well as other characteristics and parameters, enabling high-precision intelligent sensing, to provide more accurate information to the decision-making layer. Data preprocessing reduces the complexity of the data by considering only the extracted or selected data for modeling, thus improving the performance of the model [32]. The decision-making layer fuses the multi-source information, takes the multi-physical-domain fusion information as the input to the intelligent-sensing model of the thermal error of the machine tool spindle, adopts the corresponding intelligent algorithm to process to determine the estimated result of the thermal error, and analyzes the decision to detetermine the optimal strategy by considering the specific constraints [33].

### 2.3. Intelligent-Perception Model of Spindle Thermal Error Based on Multi-Source Information Fusion

By analyzing the causes of thermal error in the spindle systems of CNC machine tools, it can be seen that the primary sources of information related to thermal error are the spindle temperature field, operating condition information, and motor current signals. According to the thermal error intelligent-sensing architecture, the thermal-error sensing model of the spindle system is constructed as shown in Figure 3, including key steps such as signal acquisition, signal preprocessing, feature extraction, feature fusion, and decision fusion. In order to obtain comprehensive information about the temperature field and to avoid arranging a large number of contact temperature sensors that would interfere with normal processing, a non-contact infrared thermometer is used to obtain the temperature value of the measurement point from the thermal image. Since the spindle speed and motor current have large influences on the temperature field distribution and thermal deformation of the spindle system, the built-in speed sensor and current sensor of the machine tool are used here to obtain the spindle speed signal and motor current signal, respectively. Signal preprocessing requires filtering and denoising of multi-source signals to improve the efficiency and accuracy of the system’s signal processing. The spindle temperature field, operating condition information, and motor current associated with thermal error are extracted to construct a collection of evidence bodies in the feature space. In view of the advantages of the RBF neural network, with strong multi-dimensional nonlinear mapping ability, generalization ability, and clustering analysis ability, the RBF neural network is used to perform feature-layer fusion on the collection of evidence bodies in the feature space. At the same time, considering the uncertain information from multiple sources, the improved D-S evidence theory is further used to fuse the fusion results of the feature layer at the decision-making layer to solve the problem of accurate sensing of thermal errors of CNC machine tool spindles under complex machining environments. By weighting the fusion of different evidence, the degrees of contribution of different information sources can be more accurately reflected, making the final fusion result closer to the real situation. The specific steps are as follows:Step 1Analyze the main causes of thermal error, identify multiple sources of information associated with thermal error, and specify the type of sensor.Step 2Obtain the signal from the sensors, perform signal preprocessing, extract the feature parameters associated with the thermal error, and construct a collection of evidence bodies in the feature space to complete the training of the temperature rise–thermal error neural network model for temperature-sensitive points.Step 3The RBF neural network is used to diagnose the body of evidence in each feature space separately, and the set of preliminary diagnostic results is obtained.Step 4Calculate the basic credibility of each preliminary diagnostic result set.Step 5The reliability interval of each evidence body in the recognition framework under the action of a single evidence body is calculated according to the basic credibility assignment of the evidence body in each feature space.Step 6According to the weighted evidence fusion algorithm, the spindle thermal-error prediction value is calculated, and the thermal-error sensing result is obtained.

The image of the temperature field of the spindle system obtained by the infrared thermal imager contains a large number of temperature measuring points. In order to reduce the amount of computation associated with the sensing model and to ensure the accuracy of the sensing model, it is necessary to optimize the selection of the temperature measuring points. A combination of gray correlation and principal component analysis is used to screen the temperature sensitivities of the temperature field. Firstly, the temperature measuring points highly correlated with thermal error are roughly screened by gray correlation analysis. Then its n uncorrelated principal components are extracted by principal component analysis. The principal components are used instead of the original temperature data as inputs for the thermal-error sensing model; the steps are shown in Figure 4.

Firstly, the reference sequence in the gray correlation analysis is determined to be the thermal deformation data, and the comparison sequence is the temperature rise data of the temperature measuring points; the two sequences of data are standardized. Then, the gray comprehensive relational grade between each comparison sequence and the reference sequence is calculated, and the comparison sequence corresponding to the gray comprehensive relational grade is selected as the primary temperature rise variable, that is, the sample sequence for principal component analysis. The covariance between different factors of the sample series is calculated. Specifically, the temperature rise data of different temperature measuring points are ascertained; the contribution rates of the principal components corresponding to each characteristic root are found by calculating the characteristic root, and then selecting the first q characteristic roots and the corresponding characteristic vectors according to the contribution rate to obtain the principal components. Finally, based on the characteristic vectors, the contribution rates of the temperature measuring points as to each principal component are obtained, and the optimal temperature-sensitive points are determined.

The feature extraction of the spindle condition information needs to correspond to the sampling frequency of the temperature field to determine the speed of the spindle at the current moment, as does the signal processing of the rotational speed. Data dimension correspondence is first needed to ensure that the spindle speed sampling time is aligned with the temperature field sampling time. The data are then normalized, and the dimensionless numbers obtained from the normalization are used as features of the speed signal.

The current signal collected through RS232 is a continuous sinusoidal quantity, so the spindle motor current feature extraction must first carry out a Fourier-transform (FFT) on the continuous current flow. Spectral analysis of the current is performed to find the statistical characterization of the current signal in the frequency domain, that is, the center of gravity frequency; the calculation is shown in Equation (1). The center of gravity frequency and the effective value of the motor current are used as current signal characteristics.
(1)$$f_{c}=\frac{\sum_{i=1}^{n}f_{i}p_{i}}{\sum_{i=1}^{n}p_{i}}$$
where $q_{i}$ is the frequency domain signal of the current, the center of gravity frequency is $\;f_{c}$, and *n* is the number of points of spectral data obtained after FFT.

In order to improve the performance of the thermal-error sensing model, the temperature features, operating condition features, and current features from different data sources are fused in the feature layer to construct a collection of evidence bodies in the feature space. Then, the RBF neural network is used to initially fuse the body of evidence in each feature space separately; the specific steps are shown in Figure 5. The number of hidden nodes is first estimated based on an empirical formula, and the number of data centers is determined using the trial-and-error method. The K-means clustering algorithm determines the primary function’s data centers and expansion constants. Then, the pseudo-inverse method is utilized to calculate the output layer weights, the model error is calculated, and the training is completed after reaching the standard. When the trained RBF neural network is modeled and saved, the preliminary fusion results of a thermal error are recorded, as well as the accuracy of each neural network for decision-making layer fusion.

The RBF neural network obtains the preliminary fusion result set after fusing the feature layers of the body of evidence in each feature space, which reduces the dimensionality of the interest and extracts the effective features in it. In order to improve the accuracy of the model and make the final fusion result closer to the actual situation, this paper introduces evidence fusion at the decision-making layer; the degrees of contribution of different sources of information can be more accurately reflected through evidence fusion of different evidence.

Since the spindle mechanism speed, motor current, and temperature field of a CNC machine tool are interrelated with each other, the signals related to each thermal error cannot be independent of each other. Classical D-S evidence theory is sensitive to the basic probability assignment function and lacks robustness [34]. It suffers from fusion failure in the face of conflicting evidence. This paper employs a weighted evidence fusion theory at the decision level to address this issue. By introducing weight parameters to weight different evidence, the evidence with higher credibility or more importance has more influence in the fusion process, thus effectively solving the problem of conflicting evidence. The weighted evidence fusion is calculated as follows.

It is assumed that the set of independent possible conclusions of a problem is the identification framework $\Theta =\{A_{1},A_{2},\ldots A_{n}\}$, $A_{i}$ is the basic-element of Θ, and $2^{\Theta }$ is the power set of Θ. If the set function mapping $m:2^{\Theta }\to [0,1]$ satisfies
(2)$$\left\{\begin{array}{l} m(\emptyset )=0\\ \sum_{A\subseteq \Theta }m(A)=1,A\neq \emptyset \end{array}\right.$$
then the mapping $m:2^{\Theta }\to [0,1]$ is called the basic probability distribution function on the identification framework Θ. ∀A⊆Θ, m(A) is called the basic probability assignment of *A*. The identification framework Θ is defined by
(3)$$\left\{\begin{array}{l} Bel(A)=\sum_{B\subseteq A}m(B)\\ Pl(A)=1-Bel(A)=\sum_{B\cap A=\emptyset }m(B) \end{array}\right.$$

The mapping $\mathrm{Bel}:2^{\Theta }\to [0,1]$ is the belief function on the identification framework Θ. Mapping $\mathrm{Pl}:2^{\Theta }\to [0,1]$ is the plausibility function of Bel. For ∀A⊆Θ*,* [Bel(A),Pl(A)] is called the belief interval of A. The belief interval describes the upper and lower bounds on the degree of confidence that the current evidence holds in proposition A.

When two pieces of evidence are combined, $m_{1}$ and $m_{2}$ are the basic probability distribution functions on the same identification framework Θ. The basic-elements are $E_{1},E_{2},\ldots E_{k}$ and $F_{1},F_{2},\ldots F_{n}$, respectively. If ∀A⊆Θ and
(4)$$N=\sum_{E\cap F\neq \emptyset }m_{1}(E)\cdot m_{2}(F)>0$$
then, the synthesized basic probability distribution function $m:2^{\Theta }\to [0,1]$ is
(5)$$\left\{\begin{array}{l} m(\emptyset )=0\\ m(A)=\frac{1}{N}\sum_{E\cap F=A}m_{1}(E){\cdot m}_{2}(F),A\neq \emptyset \end{array}\right.$$

In the above formula, N is a normal number. The function is to assign the lost reliability on the empty set to the non-empty set, in proportion, to meet the requirements of the probability assignment. The N value can reflect the degree of evidence conflict; the greater the evidence conflict, the smaller the N value. The above formula is called the orthogonal sum, denoted by $m_{1}⊕m_{2}$. Moreover, the combination of evidence is independent of the order of operation. Therefore, the calculation of multiple evidence combinations can be recursively derived from the calculation of two evidence combinations to obtain
(6)$$m=\{[(m_{1}⊕m_{2})⊕m_{3}]⊕\ldots \}⊕m_{n}$$

In the weighted evidence theory, the evidence weight factor is determined by the degree to which that evidence conflicts with other evidence. For the same identification framework Θ, let the reliability of each body of evidence for identifying n propositions in the identification framework be R(A)→[0,1],∀A⊂Θ, then by
(7)$$W(A)=\frac{n\cdot R^{k}(A)}{\sum_{A\subset \Theta }R^{k}(A)}$$

The mapping *W* (·) is a weight coefficient assignment function on the identification framework Θ. *∀A ⊆ Θ*, *W(A)* is referred to as the weight coefficient assignment of the body of evidence to *A*; in the formula, and when the data on the reliability of the evidence for the identification of each proposition is more reliable, k takes a larger value. The weight coefficient *W(A)* of the evidence reflects that the evidence has different degrees of reliability in identifying the propositions in the identification framework.

The basic probability assignment function *m:*2*^Θ*→[0,1] is weighted to take full account of the weight of each piece of evidence for each proposition when the evidence is combined. ∀*A* ⊆ *Θ*, and then there are
(8)$$\left\{\begin{array}{l} W_{m}(A)=\frac{W(A)\cdot m(A)}{m(\Theta )+\sum_{A\subset \Theta }W(A)\cdot m(A)}\\ W_{m}(\Theta )=1-\sum_{A\subset \Theta }W_{m}(A) \end{array}\right.$$

The mapping $Wm:2^{\Theta }\to [0,1]$ is a weighted probability assignment function on the identification framework Θ. ∀A⊆Θ, Wm(A) is called the weighted probability assignment of A.

Therefore, the weighted synthesis rule for multiple evidence is
(9)$$Wm=\{[(Wm_{1}⊕Wm_{2})⊕Wm_{3}]⊕\ldots \}⊕Wm_{n}$$

As a result of the weighting of the basic probability assignment function, rational evidence is strengthened, irrational evidence is weakened, and conflicts between evidence are significantly reduced. Therefore, the improved weighted evidence theory can more widely meet various practical applications. The improved weighted evidence theory fully utilizes the information from the evidence sources and eliminates the incompleteness and uncertainty of the information contained in a single data source.

## 3. Thermal-Error Perception Experiment Setup

### 3.1. CNC Machine Tool Spindle Thermal-Error Perception Experiment

A large number of scholars’ experimental research has found that the spindle, in the X and Y directions, does not produce significant thermal error [35]. In contrast, the Z-direction thermal error is very obvious. So the spindle thermal error is generally established as a mapping relationship between the spindle temperature measuring points and the Z-direction thermal error. This paper’s intelligent-sensing model of thermal error mainly establishes the relationship between the temperature field at the corresponding moment, the working conditions, the spindle motor current and thermal deformation measurement point data. Therefore, the data to be measured in the thermal-error experiment are the temperature field, spindle speed, spindle current, and Z-direction thermal deformation.

The experiment takes the Yunnan Machine Tool Factory VMC850 vertical machining center as the perceived object. This paper’s experiments select an eddy current displacement sensor to measure the thermal error of the spindle system. The contact temperature sensor has many installation elements and cumbersome wiring is employed in the actual measurement, leading to inaccurate measurement results. So, in this paper, we select an infrared thermal imager to collect the temperature data from the temperature measurement points of the spindle system of the vertical machining centers. Then, the data is transmitted to the computer through USB transmission. Speed and motor current can be monitored and collected in real time by connecting the spindle’s built-in sensor to the PC through the RS232 transmission bus. The eddy current displacement sensor can realize the real-time measurement of thermal deformation of the spindle under the idle state of the machine tool through non-contact measurement. The experimental platform built for each device is shown in Figure 6.

Considering the huge structure of the machine tool and the large number of sensors required, this paper adopts an infrared imager to take pictures of the spindle system. The thermal image and its temperature measuring points are arranged as shown in Figure 7. Points a, b, c, g, h, and i are the spindle box shell temperatures; points d, e, and f are the points for the spindle motor shell temperature; points j and k are for the temperature of spindle bearing; l is the temperature of the spindle claw disc; and the m and n points will not produce noticeable temperature changes with the extension of the spindle working time, so they are used to represent the workbench and the ambient temperature.

In order to simulate the change of spindle speed during the actual machining process, the experiment was carried out in no-load mode. We edited the working procedure of the vertical machining center according to the set working conditions, collected data every 1 min, and produced four groups of data in total. Constant speed and ISO-variable-speed experimental data were used for modelling, and stepped variable speed data were used to validate the model’s predictive accuracy.

### 3.2. Experimental Conditions Design

In order to simulate the thermal drift of the spindle under actual processing conditions through no-load experiments, this project carried out thermal-error test experiments. ISO [36] testing standards identified three no-load experiments at different speeds. The temperature, spindle motor current, spindle speed, and thermal-error data were collected under different speed conditions to provide basic data for intelligent-perception modelling and model verification of thermal error. The three groups of no-load test conditions are composed of one group at constant speed conditions, and two groups at variable speed conditions. The constant speed idling was 2000 r/min for 2 h and 4000 r/min for 2 h, respectively. The two sets of variable speed conditions comprised ISO-standard variable speed conditions and stepped variable speed conditions, as shown in Figure 8 and Figure 9.

The two working conditions of constant speed and ISO-standard variable speed were used for model training, and the stepped variable speed condition was used for model testing.

## 4. Results of the Experiment and Discussion

### 4.1. Results of the Experiment and Data Analysis

The thermal characteristic experiment was carried out at the two sets of off-line speeds described in Section 3.2; the temperature rises for the measured points in the temperature field of the spindle at a constant speed of 2000 r/min, a constant speed of 4000 r/min, and a variable speed of ISO are shown in Figure 10.

The figure shows that under a constant speed of 2000 r/min, the temperature field no longer produces apparent changes after the spindle is run for 95 min. Point l (temperature measuring point of spindle claw disc) has the most drastic temperature change, with an eventual stable temperature rise of about 7.8 °C. Under the conditions of 4000 r/min constant speed, the spindle bearing and claw disc at first produced a significant temperature rise; it then dropped back to normal temperature as the cooling air conditioning started working. The spindle bearing and claw disc reached a stable temperature rise after 80 min of operation, within which the temperature rise at point l was the most obvious, rising to about 34.8 °C. After 100 min, the other temperature measuring points no longer produced significant temperature rises, and reached a stable state. Under ISO variable speed conditions, the spindle temperature field produced different temperature rises with speed, and the stable thermal equilibrium time is no longer apparent. The temperature change at point g (the temperature measuring point of the spindle bearing) is most drastic under variable speed conditions. This proves that the temperature field variation of the spindle is closely related to the speed. Between 0 and 20 min, the rotation speed of the spindle is small, and the temperatures of the relevant measuring points also change little in this period. Between 20 and 40 min, the speed of the spindle is larger, and the temperature rise of the measuring point is also larger. Between 40 and 100 min, the speed of the spindle decreases, and the temperature rise of the measuring point does not change significantly. Between 100 and 120 min, the spindle speed increases, so the measuring point has a significant temperature rise. After 120 min, the speed of the spindle decreases, and the temperature of the measuring point slowly falls back and tends to be stable.

In the off-line experiment of spindle thermal characteristics, the Z-direction thermal deformations produced by the spindle at 2000 r/min constant speed, 4000 r/min constant speed, and ISO variable speed are shown in Figure 11. As can be seen from the figure, the maximum change in thermal error when the spindle is running at constant speed occurs between 10 min and 20 min; the main reason is that the temperature of the machine tool is low after the tool is switched on, and the heat only begins to gather. The machine tool then works for a period of time, producing temperature field changes, and thus causing thermal deformation of the spindle. After 80 min, the machine reaches thermal equilibrium, and the thermal error reaches a stable value. Moreover, according to the graph for the variable speed, it can be seen that due to its lower speed, the change of thermal error in the initial state is much smaller than that in the constant speed state. It can be seen that the thermal error of the spindle is positively correlated with the speed of the spindle, and the thermal error is generated with a certain lag.

### 4.2. Comparison and Analyses of Prediction Performance of Thermal Error Intelligent Perception Models

The BP neural network is the most widely used network in thermal-error modeling, one which has good results for thermal-error prediction and is universal across different machine tools [37]. Moreover, the combination of wavelet analysis and BP neural networks can solve artificial neural networks’ slow-convergence problems and transition fitting. The wavelet neural network optimized with the genetic algorithm has higher accuracy and faster convergence. Therefore, this paper uses two thermal-error modeling methods, the traditional BP neural network and the wavelet neural network optimized by genetic algorithm, for the thermal-error comparison model.

The performance of the three models is compared for thermal-error prediction under three different operating conditions, namely, 2000 r/min constant speed, 4000 r/min constant speed, and ISO variable speed, respectively, as shown in Figure 12, Figure 13 and Figure 14. In order to verify the robustness of the model proposed in this paper, the experimental data measured under the working conditions of a stepped variable speed are input into the thermal-error prediction model as test data and compared with the measured Z-axis thermal-error, as shown in Figure 15.

As can be seen from the figure, the maximum residuals of the thermal error intelligent-sensing method based on multi-source information fusion, BP neural network and wavelet neural network optimized by genetic algorithm are 0.49 μm, 1.26 μm, and 0.96 μm, respectively, under the idling at a constant rotational speed of 2000 r/min. The maximum residuals of the thermal error intelligent-sensing method based on multi-source information fusion, BP neural network, and wavelet neural network optimized by genetic algorithm are 0.47 μm, 2.41 μm, and 0.99 μm, respectively, under idling at a constant rotational speed of 4000 r/min. The maximum residuals of thermal errors of the main shaft based on the intelligent-sensing method of thermal error with multi-source information fusion, BP neural network, and wavelet neural network optimized by genetic algorithm, under online prediction of step-variable rotational speed conditions, are 0.57 μm, 2.09 μm, and 1.52 μm, respectively. The model’s superiority under complex working conditions is further verified. The intelligent-sensing model performs well under a variety of operating conditions, proving that it is robust and able to adapt to different operating conditions and environmental changes, which reduces the need for model retraining and parameter tuning.

The results of the comparison of these three models are shown in Table 1, which com-pares and analyses the effectiveness of thermal-error prediction of the spindle under constant speed, standard variable speed, and stepped variable speed conditions. We found that the prediction accuracy of the intelligent-perception model based on multi-source in-formation fusion reaches 98.8%, which is 6.6% higher than the traditional BP neural network perception model. The traditional BP neural network has low prediction accuracy under high and variable speed conditions, is easily affected by noise and non-linearity, and has poor robustness. The intelligent-perception model based on the fusion of multi-source information improved by 4.3% over the wavelet neural network optimized by genetic algorithm. Although the optimization effect of the wavelet neural network optimized by genetic algorithm is significant, there is still room for improvement under dynamic and complex working conditions. The maximum residual errors of the thermal error intelligent-sensing model based on multi-source information fusion proposed in this paper are generally smaller than those of the traditional thermal-error prediction model. The stepped variable speed as test data in the intelligent-perception model obtained good prediction results with small maximum residual errors and high accuracy.

## 5. Conclusions

The spindle system is an essential core component of CNC machine tools, and its operational status plays a vital role in machining quality. The main idea of realizing the intelligent sensing of thermal error in the spindle system is to utilize multiple sensors to monitor the relevant information of spindle thermal errors. Through suitable mathematical models, these pieces of information are comprehensively analyzed, modeled, and processed in order to make decisions, resulting in an intelligent perception of thermal error. The decision given is the result of intelligent sensing of thermal error. The realization of intelligent spindle autonomous perception plays a vital role in its subsequent active control and predictive reasoning. This paper introduces the evidence theory, based on a neural network, establishes the intelligent-sensing model of spindle thermal error, and adopts the feature layer–decision layer approach to realize the fusion of multi-source information at different levels, which further improves the accuracy and robustness of the thermal-error sensing.

(1)Aiming at the shortcomings of single-sensor-based information, namely, characterization ability, high contingency, and susceptibility to being interfered with by external environmental factors, this paper proposes a multi-source and multi-level information fusion intelligent-sensing method. A multi-source and multi-layer thermal error intelligent-sensing model is established by feature extraction and fusion of multiple related signals affecting the thermal error of the spindle. In the model construction, the RBF neural network is used for the initial fusion of feature layers to improve the model’s generalizing ability. At the same time, the weighted evidence fusion theory is introduced, which can more accurately reflect the contribution degree of different information sources by weighted fusion of varying evidence, making the final fusion results closer to the actual value. Thus, the intelligent-perception model has a higher prediction accuracy and lays a specific theoretical foundation for developing intelligent spindles.(2)This project conducted thermal-error perception experiments on the spindle system of the VMC850 vertical machining center manufactured by Yunnan Machine Tool Factory, under the conditions of constant speed, standard variable speed, and stepped variable speed. Then we compared the effectiveness of the traditional thermal-error sensing model and the intelligent-sensing model with multi-source information fusion. The experiment’s results show that the prediction accuracy of the multi-source and multi-level information fusion intelligent-sensing model proposed in this paper can reach 98.8%, which is significantly better than the traditional model. This shows that the method proposed in this paper has significant advantages and application potential in solving the thermal-error sensing problem of the CNC machine tool spindle.(3)This paper summarizes and analyzes the signals related to thermal error inside and outside the machine tool and finally selects three signals for multi-source information fusion to obtain the thermal-error model. The amount of data is increased compared with the traditional thermal-error model. However, because the thermal-error influencing factors of the machine tool under complex working conditions are multifarious, the number and types of training samples can be gradually increased in the next study to further train and optimize the model’s parameters.(4)In addition, in order to obtain a more comprehensive thermal-error model under the various working conditions of the machine tool, it is necessary to measure the thermal error changes under different working conditions, especially under the cutting working condition. However, due to the limitations of the experimental conditions, this paper simulates the thermal error changes of the spindle under different working conditions only through the spindle’s different rotational speeds. It does not take into account the effect of the cutting heat on the thermal error. Therefore, the experiments, as to the working-condition information, can be further enriched, and a more complete thermal-error model can be obtained.

## Figures and Tables

**Figure 1 sensors-24-03614-f001:**
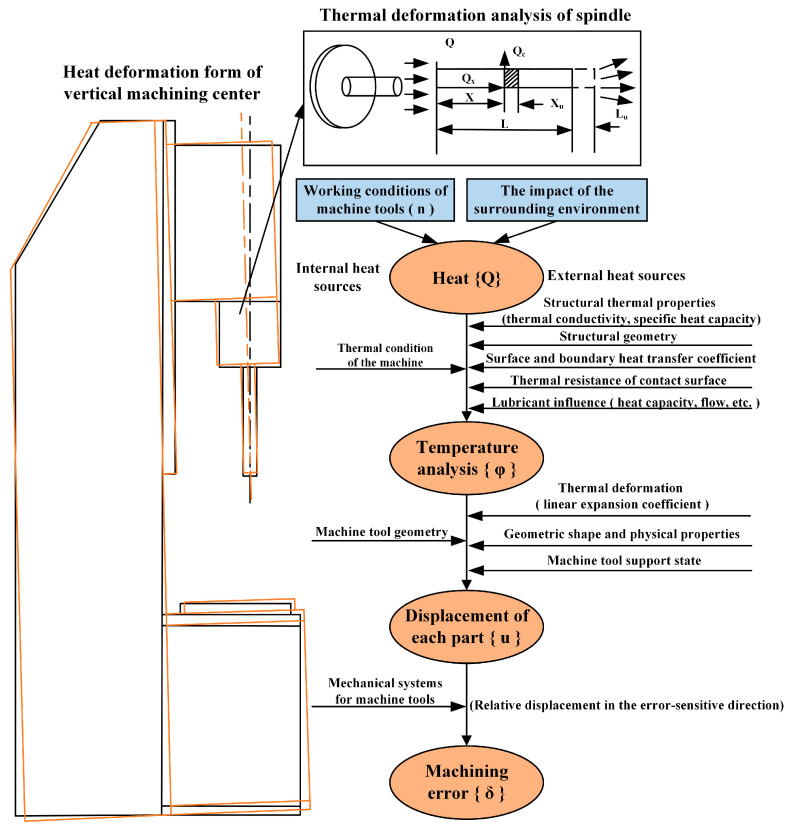
Thermal deformation mechanism of the vertical machining center.

**Figure 2 sensors-24-03614-f002:**
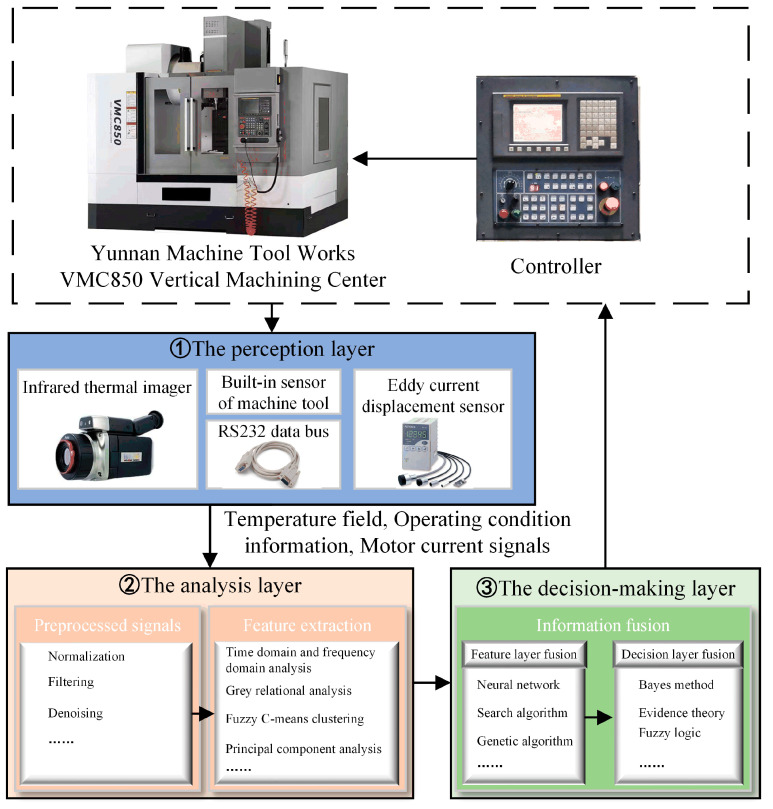
Thermal error intelligent-perception architecture.

**Figure 3 sensors-24-03614-f003:**
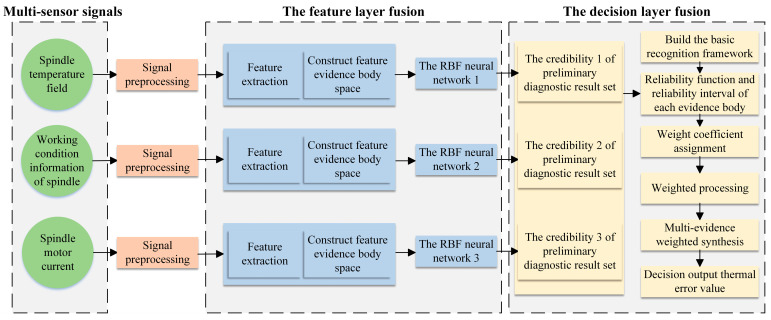
Thermal-error perception model of machine tool based on multi-source information fusion.

**Figure 4 sensors-24-03614-f004:**
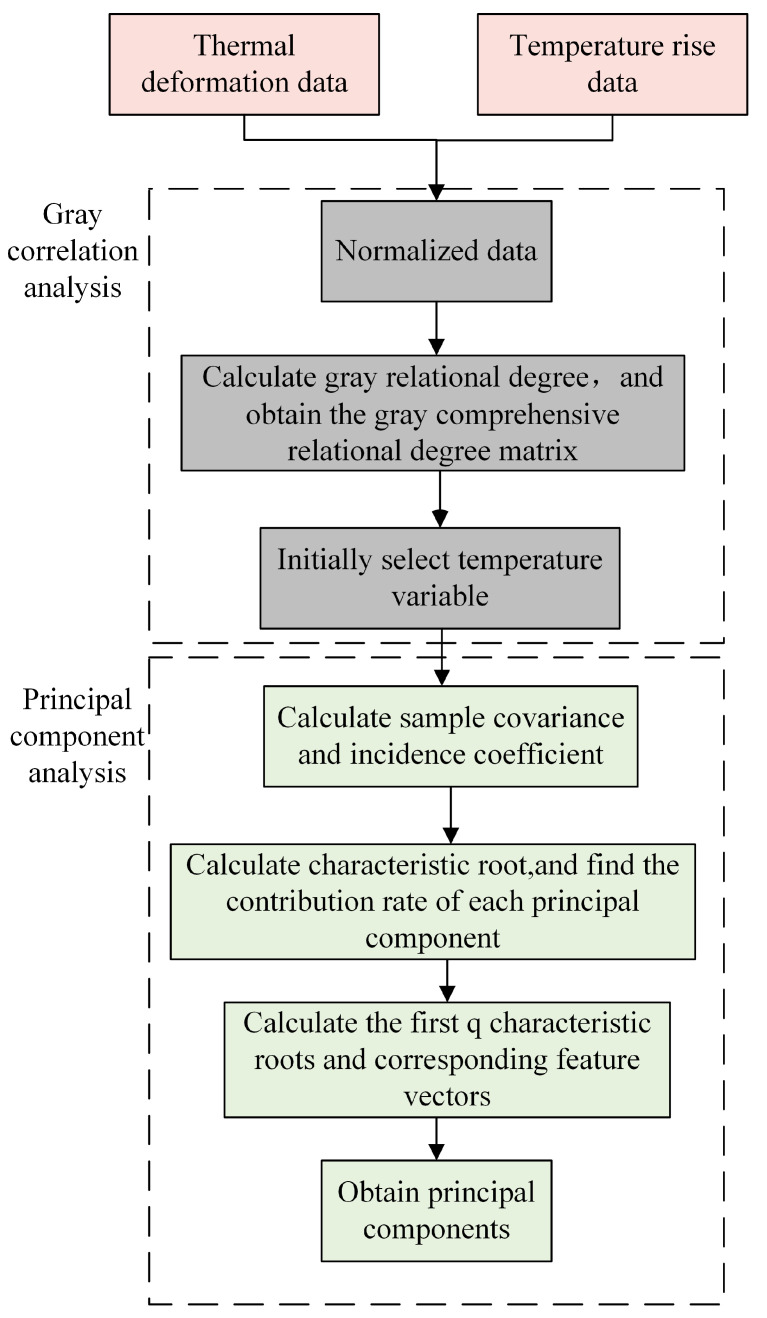
Temperature-sensitive point optimization process.

**Figure 5 sensors-24-03614-f005:**
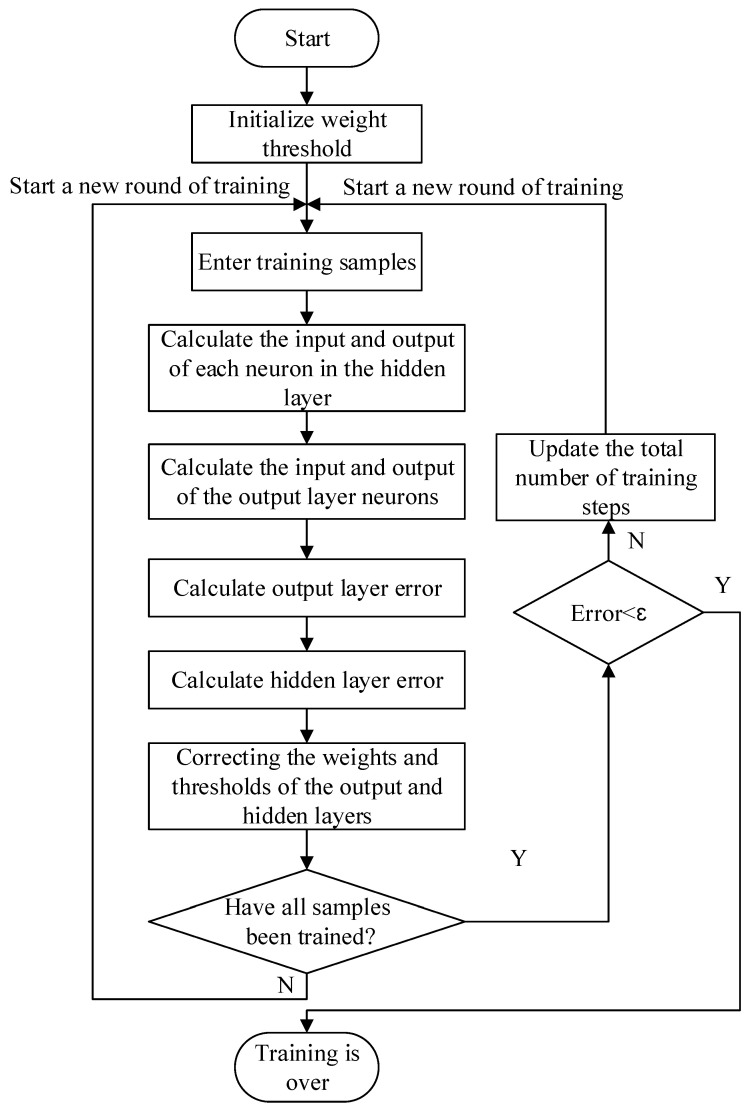
Training process of the generalized RBF neural network model.

**Figure 6 sensors-24-03614-f006:**
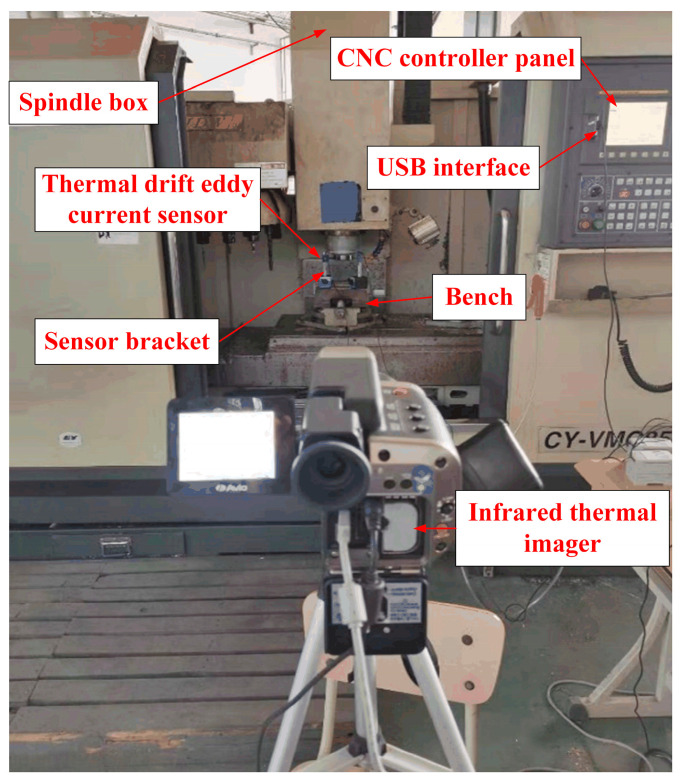
Experimental platform of thermal error intelligent-sensing system.

**Figure 7 sensors-24-03614-f007:**
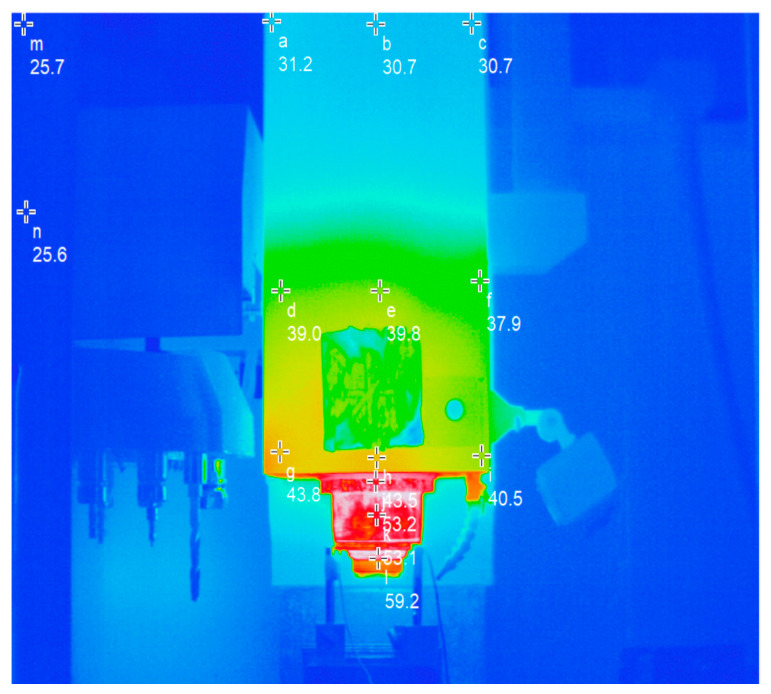
Layout of temperature measuring points at a certain time during machine operation.

**Figure 8 sensors-24-03614-f008:**
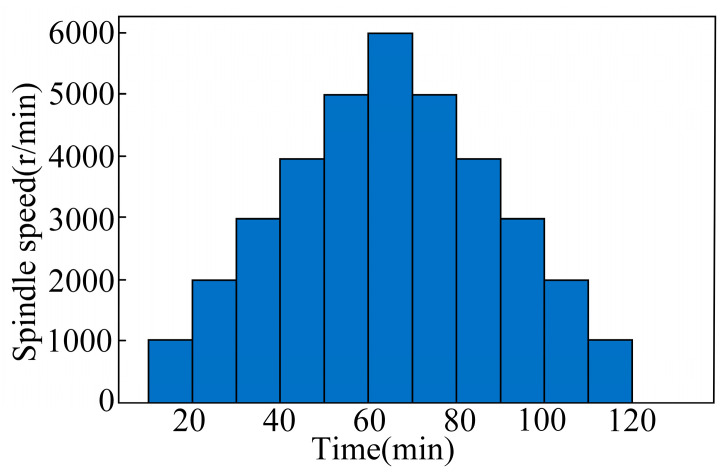
ISO-standard variable speed operation: speed–time diagram.

**Figure 9 sensors-24-03614-f009:**
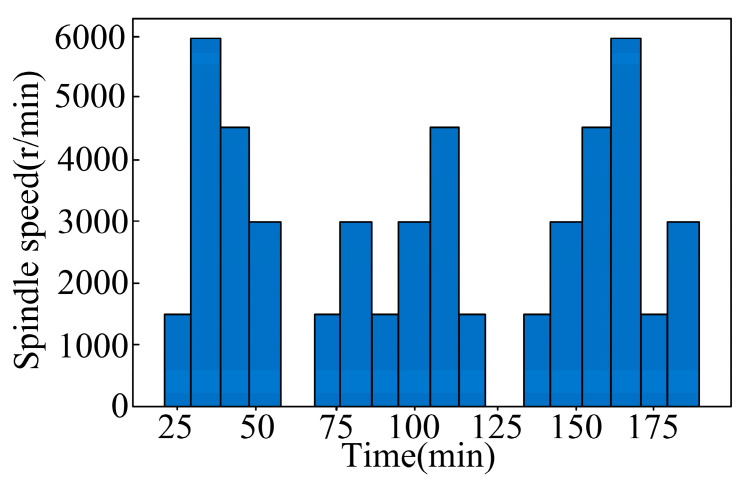
Stepped variable speed operation: speed–time diagram.

**Figure 10 sensors-24-03614-f010:**
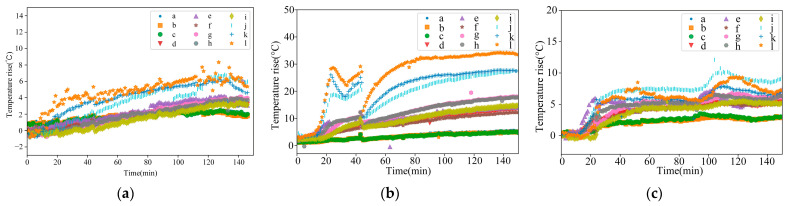
Temperature rises at temperature measuring points at 2000 r/min, 4000 r/min constant speed, and ISO-standard variable speed: (**a**) 2000 r/min constant speed; (**b**) 4000 r/min constant speed; and (**c**) ISO-standard variable speed.

**Figure 11 sensors-24-03614-f011:**
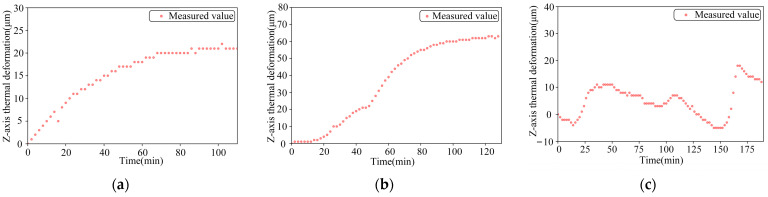
Z-axis thermal deformation at 2000 r/min constant speed, 4000 r/min constant speed, and ISO variable speed: (**a**) 2000 r/min constant speed; (**b**) 4000 r/min constant speed; and (**c**) ISO-standard variable speed.

**Figure 12 sensors-24-03614-f012:**
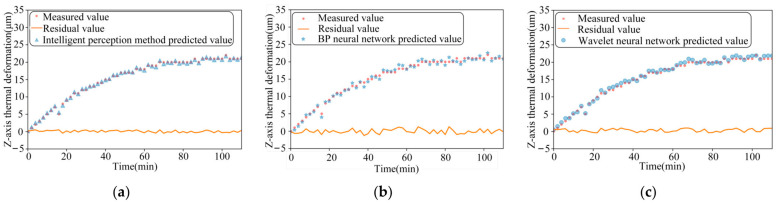
The comparison between the measured value of 2000 r/min idling and the predicted value of the three models: (**a**) Intelligent-perception method of multi-source information fusion; (**b**) BP neural network speed; and (**c**) Wavelet neural network.

**Figure 13 sensors-24-03614-f013:**
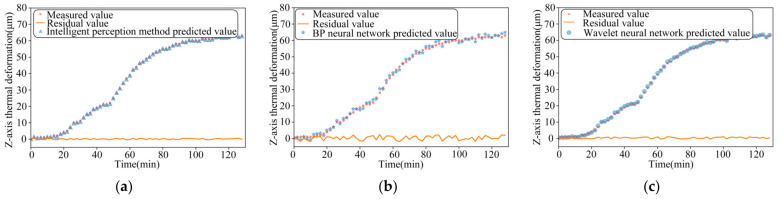
The comparison between the measured value of 4000 r/min idling and the predicted value of the three models: (**a**) Intelligent-perception method of multi-source information fusion; (**b**) BP neural network speed; and (**c**) Wavelet neural network.

**Figure 14 sensors-24-03614-f014:**
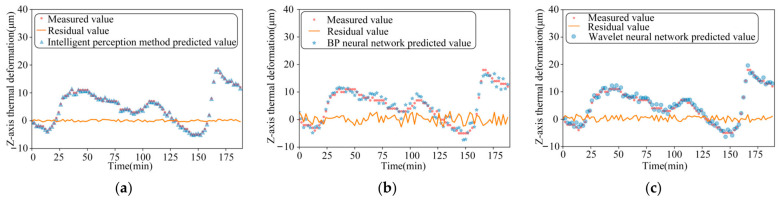
The comparison between the measured value of ISO-standard variable speed idling and the predicted value of the three models: (**a**) Intelligent-perception method of multi-source information fusion; (**b**) BP neural network speed; and (**c**) Wavelet neural network.

**Figure 15 sensors-24-03614-f015:**
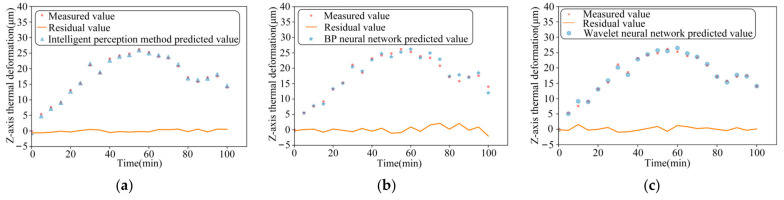
The comparison between the measured value of idling with step change speed and the predicted value of the three models: (**a**) Intelligent-perception method of multi-source information fusion; (**b**) BP Neural Network speed; (**c**) Wavelet neural network.

**Table 1 sensors-24-03614-t001:** Comparison of the prediction-effectiveness of the three models.

Model	BP Neural Network Model		Wavelet Neural Network Model		Intelligent-Perception Model	
	Maximum Residual Error (μm)	Fitting Accuracy	Maximum Residual Error (μm)	Fitting Accuracy	Maximum Residual Error (μm)	Fitting Accuracy
2000 r/min off-line detection	1.26	94.7%	0.96	96.3%	0.49	98.1%
4000 r/min off-line detection	2.41	96.2%	0.99	98.5%	0.47	99.3%
ISO variable speed off-line detection	2.97	85.2%	1.79	91.4%	0.48	98.6%
Stepped variable speed online prediction	2.09	92.6%	1.52	94.5%	0.57	98.8%

## Data Availability

Data are contained within the article.

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
