# Peer review of "Intelligent Sensing of Thermal Error of CNC Machine Tool Spindle Based on Multi-Source Information Fusion"

_sensors, 2024, doi:10.3390/s24113614_

Round 1

Reviewer 1 Report

Comments and Suggestions for Authors

Dear authors,
thank you for contributing to the multisensor data analysis using RBF-ANN analysis. I want to applaud the authors for a well-organised and well-articulated study. The manuscript is nicely constructed and provides insightful findings. However, as much as I hate to do it, I would like to provide some suggestions and recommendations, which I believe will alter the quality of the manuscript even further. My suggestions and recommendations are in the following.

Introduction

I like how you've described and provided a literature review. However, I would suggest, due to long paragraphs (lines 119-155; lines 156-193) to make smaller ones. Also, considering making subsections in the introduction that will provide more understanding to the reader. The introduction is fairly long. Also, given that you are dealing with maintenance management, I would also suggest incorporating rationale as to how does this study contributes to contemporary research the Energy-Based Maintenance and Sustainable Maintenance practice (see 10.1016/j.jclepro.2023.137177; 10.24425/mper.2020.133730). Lastly, consider adding more recent references, as I see that most of your citations are spanning from 2010-2015.

The paragraph (lines 76-86) "Regarding thermal error modeling, there are mainly thermal error modeling methods based on heat transfer theory and polynomial fitting or neural network modeling...", could benefit from more references and literature review regarding the claims presented.

Also, if you are referencing as "Huang et al. [8]..." do that throughout the whole manuscript. For instance, you've used (line 87) "KIM[7] and others...".

Place "In summary, most..." (lines 186-193) as a separate and second to last paragraph in the introduction. 

Provide a last paragraph that will describe the structure of the manuscript such as "The article is structured as follows...", and describe what have been done in each of the sections.

2. Intelligent sensing method...

Firstly, I would not recommend using the title of the section as stated. Perhaps, using Metods and Materials (or experimental design), methodology, etc., would be more appropriate in describing the contextual settings of performed experimental investigation. Also, I see that you've provided fairly significant insights about potential internal and external factors affecting the CNC machine tool spindle...but, how about the human factor? System design? Machine setup? Operations and maintenance, alignment, settings, etc. If not, it would be good to state that these factors are assumed to be..., or assumed not affecting the thermal error.

Maybe consider merging section 2 and 3 as both include description of experimental design and data (pre)processing.

A major recommendation for improvement is setting "3.3 Experimental Results and Analysis" as a Section and not a subsection as you did. Basically, this should where a section 3 should be placed. So instead of 3.3 place this as a section "3. Experimental Results and Analysis". Also, I would not advise using the term analysis but discussion = 3. Experimental Results and Discussion.

I do not know if its a mistake in a processing system in MDPI but you've stated um, I suppose micrometers?

I this (yours 4th section), I would argue that more discussion should be provided regarding the behavior of the model, fine tunning, results, implications, etc.

5. Conclusion

 A major revision should be made to the conclusion section. Namely, I would advise the authors to provide are major limitations of the study; what are exact contributions to the literature and practice; and are recommendations and future research directions from this point forward.

I hope my insights and recommendations will help you improve the quality of the manuscript. Also, I would like to thank the editor(s) for considering me as a reviewer for this article. 

Kind regards,
the reviewer.

Comments on the Quality of English Language

There are minor spelling and grammar mistakes. Authors, I suppose, mistakenly place semicolon symbols in places where I believe full stop should be placed. Major writing and spelling errors were not so significant. 

Author Response

We feel great thanks for your professional review work on our article. According to your nice suggestions, we have made extensive corrections to our previous draft, the detailed corrections are listed below.

Reviewer 2 Report

Comments and Suggestions for Authors

In this study, an intelligent perception method is proposed. This study is relatively complete. However, there are still some issues that need to be addressed.

1. The introduction of research background cannot represent the latest research progress. Please provide the latest research methods and literature.
2. Why did the experiment use an empty load to collect data? Does this match the actual processing process?
3. The temperature field is also very sensitive to environmental temperature. Is there a change in environmental temperature during the experiment?
4. How to calculate the fitting accuracy mentioned in the article?
5. In the actual machining process, there will be cutting fluid present. Is the method proposed effective in this situation?

Author Response

We feel great thanks for your professional review work on our article. According to your nice suggestions, we have made extensive corrections to our previous draft, see the word document for details.

Round 2

Reviewer 1 Report

Comments and Suggestions for Authors

After reviewing the manuscript, I believe that small changes can be made to the article. However, considering the extensive editing I do not have major suggestions for improvement. The only thing I would like to suggest is that the introduction could be reduced or at least split into smaller subsections.

The rest of the manuscript seems fine and can be suitable for publication.

Thank you for considering the changes made to the article; I wish you all the best in your future research endeavours!

Regards,
the reviewer.

Comments on the Quality of English Language

No major mistakes noticed.